# Attribution Markers and Data Mining in Art Authentication

**DOI:** 10.3390/molecules27010070

**Published:** 2021-12-23

**Authors:** Barbara I. Łydżba-Kopczyńska, Janusz Szwabiński

**Affiliations:** 1Faculty of Chemistry, University of Wrocław, F. Joliot-Curie 14, 50-383 Wroclaw, Poland; 2Laboratory of Analysis and Non-Destructive Investigation of Heritage Objects, National Museum in Kraków, 30-062 Krakow, Poland; 3Hugo Steinhaus Center, Faculty of Pure and Applied Mathematics, Wrocław University of Science and Technology, Wyb. Wyspiańskiego 27, 50-370 Wroclaw, Poland; janusz.szwabinski@pwr.edu.pl

**Keywords:** authentications, paintings, data mining, analytical procedures, forensic analysis

## Abstract

Today’s global art market is a billion-dollar business, attracting not only investors but also forgers. The high number of forged works requires reliable authentication procedures to mitigate the risk of investments. However, with the developments in the methodology, continuous time pressure and the threat of litigation, authenticating artwork is becoming increasingly complex. In this paper, we examined whether the decision process involved in the authenticity examination may be supported by machine learning algorithms. The idea is motivated by existing clinical decision support systems. We used a set of 55 artworks (including 12 forged ones) with determined attribution markers to train a decision tree model. From our preliminary results, it follows that it is a very promising technique able to support art experts. Decision trees are able to summarize the existing knowledge about all investigations and may also be used as a classifier for new paintings with known markers. However, larger datasets with artworks of known provenance are needed to build robust classification models. The method can also utilize the most important markers and, consequently, reduce the costs of investigations.

## 1. Introduction

Since early 2000s, the fine art market has made significant progress exhibiting its dynamic characters, with a 1370% turnover growth rate over a 16-year period [1]. Artprice with the collaboration of its Chinese State partner AMMA (Art Market Monitor of Artron), reported that, in 2020, the art market fell 21% down to $10.57 billion [2]. This was a surprisingly small contraction given the impact of the COVID-19 crisis on the art market as a whole (galleries, museums, fairs, etc.) [2]. Considering the above, it is not surprising that the number of fakes, forgeries and copies emerging on the art market in recent years is significant and fast growing [3,4]. Fine art authentication is a lucrative business. The opinions of well-established experts are rarely challenged [3], since forgers, by nature, prefer anonymity. However, the number of cases where confirmation of art authenticity has turned into a nightmare for an expert is on the rise. The unique story of Han van Meegeren, who is recognized as one of the most ingenious forgers of the 20th century, can be used as an example [5,6]. Just with fakes based on Vermeer, the forger amassed a fortune worth more than half a billion dollars in today’s currency. His work might have been still attributed to others if not for a cruel twist where he admitted to the forgeries in order to avoid being accused of collaborating with Nazis. History has delivers several other profiles of ingenious forgers, with the most recent one named Wolfgang Beltracchi [7]. This German painter confessed to forging hundreds of paintings in an international art scam netting millions of euros. In 2011, he was sentenced to six years in prison and, just overnight, became the celebrity called by the media “a forger of the century” [7,8].

There are two common approaches to the authenticity evaluations of paintings. The long-established, traditional one is based on the opinion of an art expert or an art historian. It is rather subjective by nature, and as such, it sometimes does not comply with the demands of a modern forgery detection procedure. However, the beginning of the 20th century witnessed many spectacular scientific and technological developments, a wide range of which turned out to be useful for forgery detection. In the 1930s, for instance, basic analytical investigations, including X-ray [9] and UV photography [10], as well as a chemical analysis of the pigments [10], were introduced to fine art authentication. Their application in the analysis of the painting materials (pigments, dyes, binding media and support) has aided experts in confirmation of the authenticity of analyzed pieces of art and verification of the consistency of the materials used in the objects in question while keeping in mind the supposed time of its creations or authorship. Since the scientific approach is more objective than the traditional one, including comprehensive physicochemical examinations into the evaluations of paintings has become a necessity in the last decades.

Art experts, scientists or other experts involved in authenticity trials often find that terms like “highly probable”, “suggested” and “possible” widely used in the scientific literature are not satisfactory for the court. In most cases, the expert is under pressure to answer the question about authenticity in the simplest “yes” or “no” terms. Sometimes, the court will accept evaluations delivered in percentages; for example: “the object is 80% authentic”. Situations where it is possible to deliver a “yes” or “no” answer are the easiest, but how to quantify authenticity in numbers when the final opinion is indecisive? The combined data gathered during the case study might be strongly suggestive, but still, an answer totally verifying the attribution of the objects is troublesome. In such cases, an objective procedure based on the results obtained from a comprehensive investigation might be advantageous.

In this paper, we examine whether the decision process involved in authenticity evaluations may be supported by machine learning algorithms. The concept of decision support by a mathematical algorithm is not a new one. It has evolved from the theoretical studies of organizational decision making performed in the 1950s and 1960s [11]. Although the decision-making systems were intended to support the processes in management, in recent decades, they have become popular in other fields, such as healthcare or the service sector. For instance, many clinical decision support systems are already used by practitioners in diagnostics, drug dosing and drug prescribing [12]. Scoring systems are also being deployed to help banks in decisions of whether to grant credit [13].

Due to the vast amount of data collected in almost every area of our lives, data-driven decisions or predictions are becoming more and more popular. The term “machine learning” was coined in 1959 by A. L. Samuel [14] and is understood as giving computers the ability to learn without being explicitly programmed.

Machine learning algorithms can learn from and make predictions about data. One of the typical tasks tackled by them is classification. Provided the existing inputs may be divided into two or more classes, a classification algorithm produces a model that assigns unseen input to one or more of those classes [15]. An example of such a classification task would be assigning a given email to “spam” or “not spam” categories based on its contents [16]. However, applications are not limited to the IT area. A diagnosis could be, for instance, assigned to a patient when described by some observed characteristics [12]. A customer’s history may indicate whether he or she is inclined to buy a product or not [17,18]. Some patterns discovered in the data may indicate fraudulent attempts in many fields (e.g., telecommunications and taxes) [17,18]. Assuming that data on already conducted authenticity analyses of paintings is available together with their outcomes, from the conceptional point of view, the decision on the authenticity of a new piece of art is nothing but a classification task. Thus, at least theoretically, one could think of a system summarizing all existing data on that painting and suggesting a category it belongs to (e.g., “authentic” or “not authentic”) on the basis of a comparison between the known cases and the one of interest.

Machine learning relies on computational statistics and mathematical optimization. As such, it often requires some mathematical background to interpret its results. Consequently, some reluctance against them may be observed among nonexperts. However, there is one concept called the “decision tree”, which has a tremendous potential as a decision supporting tool [19], mainly due to its comprehensibility. Decision trees are simple to understand and to interpret, even by people with no prior expert knowledge. They may be visualized with graph-like structures easily translated into a set of readable rules of the form “if condition 1 and condition 2 and condition 3 are met, then”.

Once the representation is generated, one does not even need a computer to apply it. That is the reason why we decided to use decision trees for our purposes and check their applicability in art forgery detection. The main goal of this paper is therefore to assess if machine learning algorithms may indeed be used as a kind of support for art experts.

We would like to combine decision trees with the attribution markers of paintings. The latter may be seen as outcomes of different analytical methods, both scientific and based on opinions, applied to the paintings under investigation. They have been identified in an in-depth analysis of over 50 case studies (Appendix A) as factors that strongly influence the decision process in forgery detection. Every painting may be characterized by a vector of the markers that summarize the most important information regarding that piece of art. Those vectors will then be used as inputs for the decision tree model we decided to use in this work.

## 2. Materials and Methods

### 2.1. Statement of Authenticity

Authentication of a piece of art even nowadays is often based solely on the opinion of an art expert or an art historian. However, based on market demand and the currently available tools, a real statement of authenticity should include at least three levels of analysis (Figure 1): confirmation of provenance (i.e., authenticity of documents supporting ownership), verification of the artistic style of work by an art expert and scientific analysis of the object using generally approved methods, both invasive and noninvasive. The proposed authentication scheme matches the forensic analysis of various documents that is already well-established [20,21,22]. An in-depth analysis of the documentation associated with the art in question can assist in formulation of the final assessment. Scientific examination of the piece of art in question incorporates investigations of painting materials like pigments and media, canvas, wooden support, nails, etc. Discovery of the materials inconsistent with the supposed time of creation would suggest that the analyzed object is not authentic. On the other hand, the consistency of all the materials employed in the investigated piece of art with the time of creation does not necessarily prove the attribution of that object. In such a situation, further analysis might be required.

The scientific analysis (i.e., the last step in Figure 1) consists of a wide range of various noninvasive and invasive analytical methods. In our opinion, the authentication process can benefit from purpose-oriented organization and implementation of these methods. A scheme presented in Figure 2 summarizes the proposed general approach applied in such tasks. The individual steps refer to the types of investigations and to the particular materials that should the analyzed but not directly to specific analytical techniques. The proposed scheme should also cover the procedures of collecting, storing and protecting samples, as well as general analysis methods. Moreover, the scheme should be repeatable and applicable to various objects. It includes various possibilities that should be considered during investigations. However, new investigative paths can be suggested with the growing amount of studies.

### 2.2. Case Studies

The analytical procedure outlined in Figure 2 has been applied in more than 50 case studies (Appendix A). The investigated collection of paintings originated from the 15th to 20th centuries. Some of the them were already positively attributed to J. M. Willmann [23,24,25,26,27], J. J. Kniechtl [28,29], A. Grottger [30], G. Penni (an apprentice of Raphael) [31], El Greco [32,33] or to the workshop of H. Bosch [34]. In addition to the collection of paintings, the reported study included a unique set of historical maps [35]. The studies were conducted by the Cultural Heritage Research Laboratory at the University of Wrocław and the Laboratory of Analysis and Non-Destructive Investigation of Heritage Objects (LANBOZ) in Krakow [36,37,38]. All of the analyzed objects were subjected to combined spectroscopic analyses that included noninvasive investigations (Vis photography, UV fluorescence, IR photography and reflectography, X-ray photography and false color analysis,); X-ray fluorescence (XRF); macro-X-ray fluorescence (MAXRF); optical coherent tomography (OCT); optical microscopy of cross-sections (MO); scanning electron microscopy with energy dispersive spectroscopy (SEM-EDS); FTIR spectroscopy; micro-attenuated total reflection (ATR) techniques; micro-Raman spectroscopy and gas chromatography with mass spectrometry (GC-MS) [39].

The proposed 3-step authentication model (see Figure 1) has been applied, for example, to old-printed maps and a painting by J. J. Knechtl. As far as the maps are concerned, their prices offered on the antiquarian market are substantially higher when the objects on sale are colored rather than black and white. Marketability “enhancement” achieved by the addition of colors to maps printed between the 16th and 18th centuries is one of the most commonly seen fraudulent alterations. Step one of the proposed process was satisfied by the fact that the authenticity of the investigated maps was not questioned; they were printed, on paper, between the 16th and 18th centuries. The question was if the genuine maps were black and white or whether they were indeed printed in color. The documentation associated with the maps was authentic; nevertheless, the art experts could not confidently assess the authenticity of the objects, which led to step three, physicochemical studies to provide a more in-depth evaluation. As can be seen in Figure 3 (left panel), several noninvasive techniques were applied to assess if the maps were originally printed in color or if the color was added to enhance their market value: XRF, micro-Raman, fiber optic mid-FTIR and near-FTIR, UV/Vis fluorescence and UV/Vis absorbance [35].

In the second case study (Figure 3, right diagram), the proposed 3-step authentication process was applied to one of the paintings that were recently discovered in a private collection in Poland. The set was attributed to Joseph Jeremias Knechtl, one of the most famous painters of the 18th century in Silesia (Poland) who was nearly completely forgotten in the following centuries. While additional documentation associated with the painting was not available, the art historian was fairly confident that all but one of the discovered paintings were indeed authored by J.J. Knechtl. Since the first two steps of the process could not confirm the attribution of the painting called “Bolko II Świdnicki”, a comprehensive comparative study implementing microscopic and spectroscopic analytical techniques was performed in order to authenticate. For the study, the art historian selected paintings representing all periods of Knechtl’s creativity. They were subjected to noninvasive analyses like X-ray; IR reflectography; UV fluorescence and complementary examinations using micro-Raman spectroscopy, micro-ART spectroscopy, HPLC (high-pressure liquid chromatography), optical microscopy and SEM-EDS examinations. The information obtained for the representative set selected by the art historian allowed the creation of a “database” of painting materials used by Knechtl. The database was then used as a validation tool for investigation of the painting whose authenticity was questioned.

### 2.3. Attribution Markers

An in-depth analysis of over 50 case studies (Figure 4, for some examples) devoted to attribution and authenticity investigations of paintings allowed us to identify several markers [40] that strongly influenced the decision process. They are listed in Table 1. The application of attribution markers in authenticity assessments is fairly new, and many of the markers need detailed explanations, since their meanings are not well-established yet. For example, a marker called “verification of the artistic style” means that an art historian performed a robust analysis of the artistic style of the painting in question. ”Historical pigments” indicates that pigments and dyes identified in the analyzed piece of art were consistent with the supposed time of creation of the investigated object. ”Stratigraphy” implies that the stratigraphy of the sample collected from the analyzed painting was compared to the samples originating from a painting of known provenance and attribution. ”Distinctive value”, also referred to a “fingerprint” of the painter, refers to the unique feature of the painting technique or the specific material used by the painter. The marker “accessory minerals” attests to the origin of the raw materials used in the ground layer [29]. One of the markers listed in Table 1 and called “state consistent” is highly unique, since it is not determined by any analytical technique. It indicates the overall consistency of all available data, like materials, technique, preservation, etc., with the known facts about the expected creator. For example, the gathered data have not delivered any information contradicting the authenticity, but the craquelure does not look like the original or the varnish seems to be quite fresh, regardless of exhibiting the expected type of fluorescence. A similar factor is used in the assisted diagnostics [41].

### 2.4. Analyzed Dataset

The data obtained during the scientific investigation of 55 paintings were used as a training set to confirm the applicability of the data mining process guided by decision trees for art classification. All the paintings were characterized by a series of measurements corresponding to the markers (Table 1) appropriate for each specific case study. The values of the markers are shown in Table 2 and Table 3. Each column in these tables represents a single painting and delineates the scientific data (markers) available for the investigated painting. The numbers 1 and −1, represent measurements with positive and doubtful/negative outcomes, respectively. An empty cell means that a factor was not investigated for the given painting.

### 2.5. Decision Trees

Decision trees are a simple machine learning algorithm that illustrates how a target variable (authenticity assessment, in our case) is explained or predicted using a set of predictor attributes (i.e., attribution markers) [42]. As their name conveys, decision trees are tree-like diagrams composed of nodes. The topmost node, called the root, represents the whole dataset. It is split into child nodes by a selected attribute to produce subsets of data with smaller impurities than the original one. The procedure is then repeated until the nodes cannot be further partitioned (because they contain data samples of the same kind, e.g., authentic paintings only) or the maximum depth of the tree is reached. In the latter case, no perfect partitioning of the data has been reached.

Every internal node of the tree represents a test on a marker. Each branch is an outcome of the test (see Figure 5 and Figure 6). The paths from the root to leaf nodes, i.e., the nodes with no children, serve as the classification rules.

The biggest advantage of the decision trees is that they are very simple models. They require little statistical background, are easy to interpret and are very for in conveying information. As for the drawbacks, due to a tendency toward overfitting, they are usually not the best choice for a robust classifier. Having that said that, we will stick to decision trees for their interpretability.

When applied to attribution markers, the decision tree method should assist in: (a) determining which markers are necessary, (b) establishing their relative relevance and (c) predicting whether a painting with known values of the attribute markers is authentic. The decision tree approach should also improve the reliability of partial authenticity assessments.

## 3. Results and Discussion

The open-source programming language Python [43], together with the Pandas and Scikit-learn [44] modules, were used to analyze the data and to build the decision tree model.

### 3.1. Dataset Characteristics

Our dataset consisted of 55 paintings, 43 of which were assessed as authentic (see Table 2 and Table 3 for details). The investigations performed for each item are shown in Figure 7 (to recall, each investigation corresponds to one attribution marker). The paintings differed from each other in the number and type of tests conducted to check their authenticity. In other words, there was no standard set of attributes the examining laboratories checked during the authenticity assessment. In fact, as is shown in Figure 5 and Table 2 and Table 3, the presented dataset did not include any sample with all 32 markers collected.

Usually, less than 20 tests are performed to classify a painting. The reason is at least twofold. First, many case studies are subjected to several limitations, including time and/or budget constraints, restricted access to a specialized equipment or lacking collaboration between experts from different domains. Moreover, the individual investigations are often conducted sequentially, one after the other. It may happen, in this case, that a subset of them is already conclusive, making the rest superfluous.

To elaborate more on that issue, let us have a look at how often every attribution marker was used. The results of the assessment are shown in Figure 8 and indicate that, indeed, there is no standard procedure for verifying authenticity. The usage frequencies for the analyzed markers were spread over a broad range of values. Additionally, five attribution markers (“unconfirmed ownership”, “test of authenticity”, “support dendrochronology”, “nails composition” and “original varnish”) were not used at all. It seems that they were considered not crucial, at least for the available dataset, and might probably be omitted in further considerations. In the literature, they are rather rarely listed in authenticity investigations [6].

In Figure 9, the investigations with negative outcomes are shown. The analyzed dataset suggests some level of differentiation in the importance/contribution of given authenticity marker to the overall investigation. Three out of four markers with the highest negative outcome were associated with the pigment analysis (“dating_pigment”, “pigments_characteristic” and “historical_pigments”). Such results underline the significance of the pigment analysis in the overall authentication process. In the case of the “dating_pigment” marker, any negligence in identifying all the places in which conservation work was performed can result in erroneous authenticity attribution. The highest number of negative outcomes was assigned to the “state _consistent” marker. In the next section, this marker is discussed in more detail. Nevertheless, the attribution markers shown in the figure are of great importance. Since one expects a painting with all positive examinations to be authentic, the forged category has to be related to negative values of the markers.

### 3.2. Building the Decision Tree

In our first attempt to build a decision tree, we split the data shown in Table 2 and Table 3 into two subsets: the training one consisting of 46 paintings (with IDs from 0 to 45) used to train the model and the test one with the remaining nine paintings. The latter one was then used to assess the accuracy of the resulting classifiers. This partitioning of the data is in accordance with its collection chronology. At the beginning of the project, only the data for the first 46 paintings from Table 2 and Table 3 were available. The remaining part was provided much later. Thus, it was a rather natural choice to use them for the test purposes. It should be stressed, however, that such a manual splitting of data is not the usual approach to machine learning. We will address that issue later in this section.

Our training set contained 36 authentic paintings and 10 forged ones (see Table 2 and Table 3). The DecisionTreeClassifier object from the scikit-learn library [44] was used to train the model. We used trees with a maximum depth equal to four. Entropy [19] was used to measure the quality of the split at each node of the tree. A search of the parameter spaces of the model was performed in order to choose the above values.

The resulting decision tree is shown in Figure 5. We see that the splits determined by the algorithm are clearly outlined and easy to trace. As already mentioned, we started with 46 samples at the root node. The “state consistent” marker was used for the first split. The test on the marker at the root may indicate that we are dealing with continuous values, but it is only an artifact of the visual representation of the trees. In our case, the condition state consistent ≤ 0.5 is satisfied if the value of the marker is simply −1 (i.e., a negative outcome of the corresponding investigation) or 0 (missing value, no corresponding investigation). The value of entropy at the root (0.755) indicated that the sample contained a mixture of authentic and forged items. The data was split into two child nodes with 11 (left) and 35 (right) samples. The left child corresponded to paintings satisfying the condition at the root node. Note that, after the split, the right child was already pure (entropy equal to 0) and contained authentic paintings only. Among the 11 paintings in the left node, one was authentic, and the rest were forged. To separate them from each other, two further splits on the left node were carried out.

First, the supporting documents marker was used to split the subset into two child nodes. Again, the left one corresponded to negative or missing values of the marker. The second split in the left node, based on the supporting documents, created two child subsets. The left child consisted of nine samples and was already pure. In other words, all the samples with both state consistent and supporting documents markers with values −1 and 0 were forged. The right child consisted of only two samples, one authentic and one not, which could be separated from each other by the application of yet another marker, the typical ground layer.

The decision tree applied in the reported case studies may be translated into the following set of rules shown in Algorithm 1:
**Algorithm 1.**if state_consistent equal to −1 or 0:   if supporting_documents equal to −1 or 0:      return FORGED (9 paintings)   else:      if typical_ground_layer equal to −1 or 0:        return AUTHENTIC (1 painting)      else:        return FORGED (1 painting)else:   return AUTHENTIC (35 paintings)

To conclude this section, we would like to stress that continuing the partitioning of data until all of the leaf nodes contain pure samples is fine if the major goal of the procedure is to summarize and convey the information contained in the data. However, it is not the best idea if one looks for a reasonable classifier, since, in this scenario, the model will probably overfit and have some problems with the generalization of unseen data. In this case, it would be probably better to stop after the first split.

### 3.3. Relevance of the Attribution Markers

The decision tree shown in Figure 5 suggests that the “state consistent” marker is the most important one in the authenticity attribution cases. It is not only optimal for the first split but also allows partitioning of the data into a pure subset of authentic paintings and an almost pure one with all but one of samples being forged. The other two attributes shown in the tree are used to extract the remaining authentic painting from the latter subset.

Surprisingly, based on the presented case study, it seems that only a small subset of the proposed attribution markers is required to assess the authenticity of the paintings. Among them, as it was already shown, the “state consistent” marker is of great importance. This attribute is often underestimated or even disregarded by experts involved in physicochemical investigations, because it does not describe any analytical results. Rather, it stands for an overall impression of an art expert. Interestingly, this rather subjective marker seems to be very important in an art historian’s expertise.

A conservation scientist examining the authenticity of a painting relies on the results obtained via application of a broad range of well-established scientific techniques. While the results might not be 100% conclusive, the analysis does not rely on emotions and, hence, can be perceived as more objective than the opinion of an art expert. However, by definition, the “state consistent” marker is a subjective feeling- and experience-based assessment of art experts. This subjective impression that “something is not right” can, and often does, lead to additional scientific investigations that would have not been performed if the art expert was able to form a firm opinion.

The relevance of a marker in the decision tree may be estimated by the so-called feature importance. It is calculated as the (normalized) total reduction of the entropy brought about by that marker. If the reduction is very close to zero (referred to as vanishing importance), the corresponding marker is irrelevant and may be omitted in the analysis.

The attribution markers with nonvanishing importance are listed in Table 4. The values confirm our previous findings. The “state_consistent”, “supporting_documents” and “typical_ground_layer” features present themselves as the most important factors. The remaining ones can be neglected.

### 3.4. First Attempt to Classification

Let us examine the efficiency of the decision tree model when it is applied as a classifier to new paintings. As it was mentioned before, the test set consisted of nine samples, seven of which had been assessed by experts as authentic. The attribution markers for those paintings are summarized in Table 3 (IDs from 46 to 55). The results of the classification are presented in Table 5. As we can see, the classifier made one mistake by predicting the “Authentic” class for one of the forged paintings. In other words, the accuracy of the classifier was equal to 88%. In general, accuracies close to 90 percent are perceived as good.

### 3.5. Decision Trees on Subsets of Features

The decision trees can also be used to check the consequences of feature removal from the training set. For instance, assume that there is no “state consistent” marker in the data or that this subjective marker has been intentionally removed by an expert. The tree built for that case is shown in Figure 6. The new model requires more features to separate both classes. However, it is still possible to achieve full separation with a rather short tree. The marker called “pigments characteristic” becomes the most important one and allows for a division of the painting into two child nodes with 10 (two authentic and eight forged) and 36 (34 authentic and two forged) samples. The importance of all markers used in the tree are summarized in Table 6. Again, the revised model displays 88% accuracy in predicting a painting’s authenticity when applied to the test data.

### 3.6. Cross-Validation of the Classifier

The overall accuracy of 88% is not bad, but until now, we worked with the same data split all of the time. Since the classification results can depend on a particular choice for the training and test subsets, we carried out a stratified k-fold cross-validation of the model. In this procedure, the data is split randomly into k smaller sets (called folds) in such a way that the percentage of samples for each class is preserved across the folds. Then, a decision tree is built using k-1 folds as the training data. The remaining fold is used for validation. The procedure is repeated for different partitionings of the folds into the training and test subsets. The final performance is simply an average of the values computed for each split.

For the data containing the whole marker set, the average accuracy of the classifier is 89% in the case of the five-fold cross-validation and 87% for the three-fold one. After the removal of the “state consistent” marker, we achieved 93% and 87%, respectively. The most important features turned out to be the same in all iterations of the validation procedure. However, their relative importance was slightly different from those presented in Table 4 and Table 6.

### 3.7. Towards a Robust Classifier

From the above results, it follows that decision trees may indeed be used as a tool supporting experts in the process of art authentication. First of all, they may help to summarize the partial outcomes of individual investigation steps. Moreover, they allow for identifying the most significant attribution markers based on the available data. This is of particular importance if there are some budget or time constraints for the analysis, since the experts may then focus only on the decisive investigation procedures.

The preliminary results for classification are also very promising, despite the fact that we used one of the simplest methods among the available algorithms, and we did not put a lot of effort into its optimization. The overall accuracy close to 90% was very good. However, a closer look at the predictions of the classifier (Table 5) revealed that its recall, which measured the fraction of important instances among the retrieved ones, was low for forged paintings. This was simply due to the fact that our dataset was unbalanced, with much more instances of authentic paintings.

Larger datasets with more balance between the classes are required to build better classifiers. However, then one would probably go for more advanced classification algorithms known to have better performance than these decision trees (e.g., random forests or neural networks; see Reference [45] for review).

## 4. Conclusions

In this paper, we studied the applicability of a method combining attribution markers [39] with the decision trees [19] for the authenticity assessments of paintings. Decision trees are a machine learning algorithm, which is very easy to interpret even by people with no prior expert knowledge from the IT and math domains. Due to this, they are often used as a decision supporting tool. When applied to attribution markers, decision trees should assist in: (a) the identification of the important markers, (b) establishing the relative relevance of the markers and (c) a prediction of authenticity of a new painting with known values of the markers. From our analysis, it follows that decision trees indeed meet all of our expectations and may be used as a tool supporting art experts.

With the help of a decision tree trained on a subset of 46 paintings (including 36 authentic ones), we were able to identify the important attribution markers and to establish their rankings. Interestingly, the “state consistent” marker, indicating the overall impression of an art expert about the painting, turned out to be the most important one. This finding is in line with the results in the assisted diagnostics [45].

The resulting decision tree was also checked as a classifier; it was able to identify the authenticity of the paintings in the test set (nine paintings—among them, two forged) with an accuracy of 88%. However, even if one does not trust the classification capabilities of the algorithm, the method can still be used to summarize the available data. Having a summary in the form of an easy-to-read tree-like structure, the actual decision on an investigated painting should be much easier to accomplish.

As already mentioned in the previous section, larger sets of good quality training data will be required for the development of more robust classifiers.

## Figures and Tables

**Figure 1 molecules-27-00070-f001:**
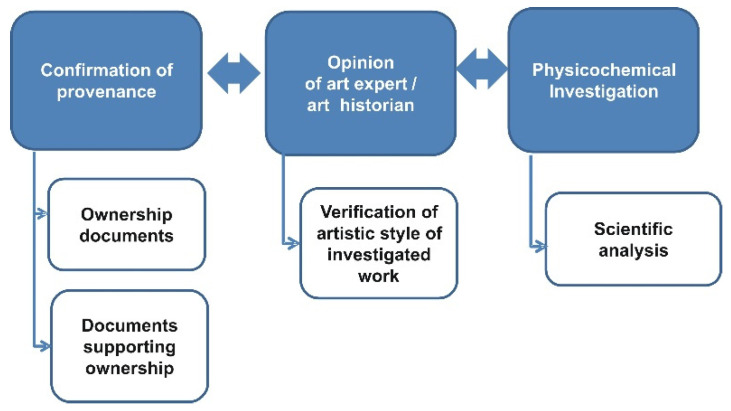
Three levels of the modern authentication procedure. The proposed scheme matches the forensic analysis of various documents that is already well-established.

**Figure 2 molecules-27-00070-f002:**
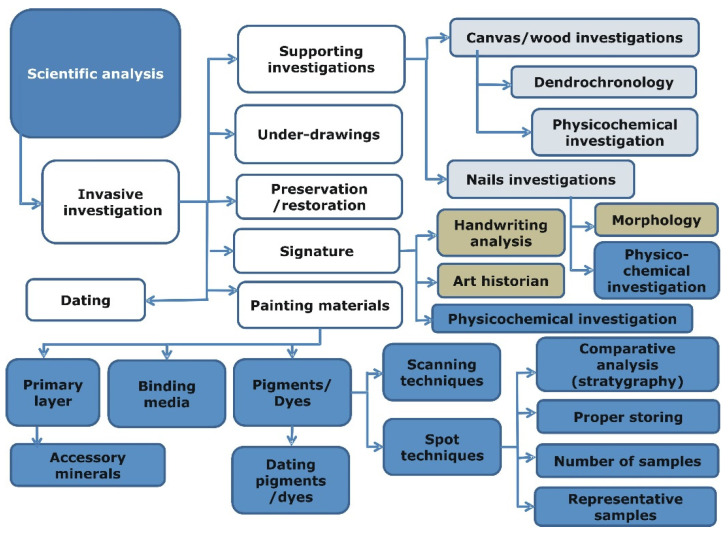
Analytical procedures supporting the authenticity examination process. Components of the scientific analysis (the last step of the procedure shown in Figure 1). The individual elements of the diagram correspond to different types of investigations.

**Figure 3 molecules-27-00070-f003:**
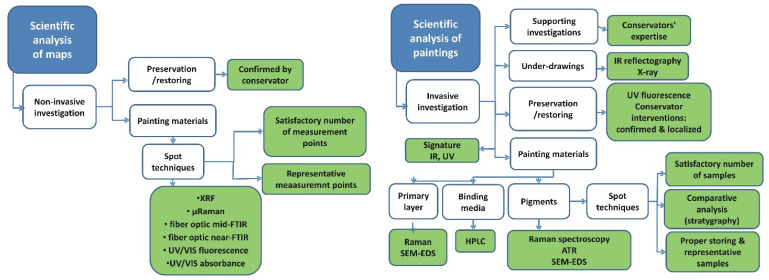
The proposed authentication model applied to different cultural heritage objects: maps printed between the 16th and 18th centuries (left panel) and the “Bolko Świdnicki” painting by J.J. Knechtl (right panel). Specific analytic techniques are now assigned to different types of investigations (see Figure 2 for the definitions of the latter ones).

**Figure 4 molecules-27-00070-f004:**
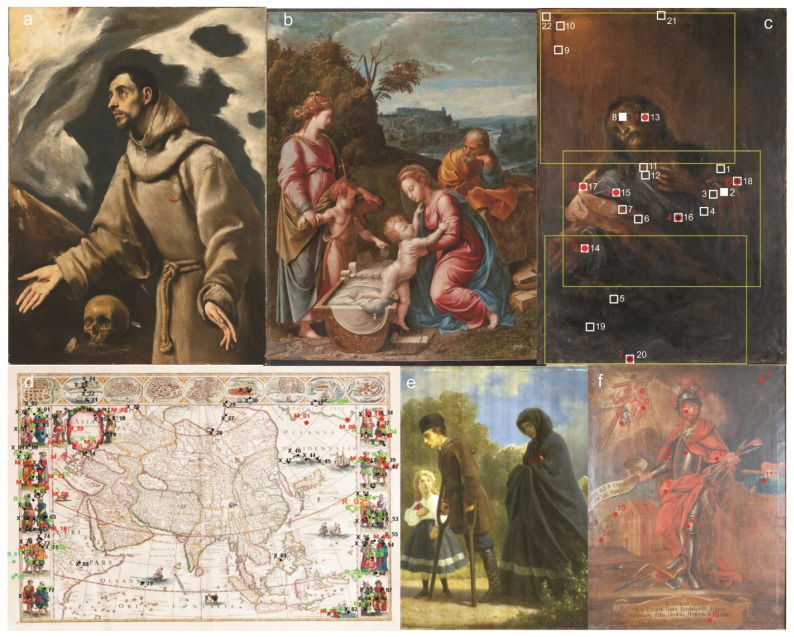
Example of investigated pieces of art: (**a**) El Greco “Ecstasy of Saint Francis” ca 1575–1580, the Diocese Museum in Siedlce (Poland). (**b**) G. Penni “The Holy Family with Saint John and Saint Catherine”, National Museum in Warsaw (Poland). (**c**) M.L. Willmann “Christ in Gethsemane”, Church of the Assumption of the Blessed Virgin Mary in Żagań (Poland). (**d**) Willem & Joan Blaeu, Theatrum orbis terrarum sive Atlas Novus, Amsterdam 1649–1655, Ossoliński National Institute in Wrocław (Poland). (**e**) A. Grotter “After the uprising” 1864, National Museum in Wrocław (Poland), (**f**) J.J Knechtl “Bolko II Świdnicki” ca. 1720, Krzeszów (Poland).

**Figure 5 molecules-27-00070-f005:**
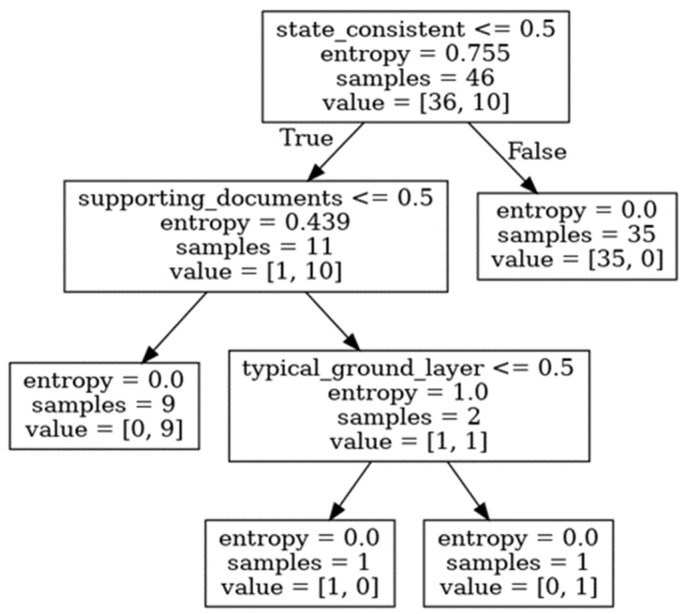
Decision trees built on the training data (i.e., paintings with IDs from 0 to 45 in Table 2 and Table 3). The “state consistent” marker seems to be the most important one for splitting the data, since it allows dividing it into a pure sample of authentic paintings (right child) and a sample consisting of all but one of the forged paintings (left child). Two further splits were performed to separate the one authentic painting from the forged sample.

**Figure 6 molecules-27-00070-f006:**
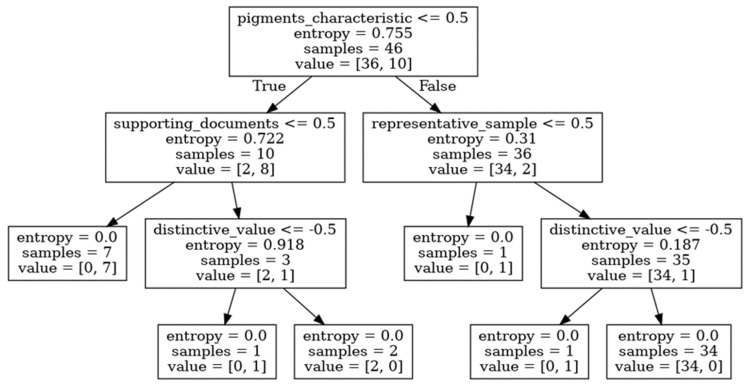
Decision tree built after the removal of the “state consistent” marker from the dataset. The “pigments characteristic” feature has become the most important one.

**Figure 7 molecules-27-00070-f007:**
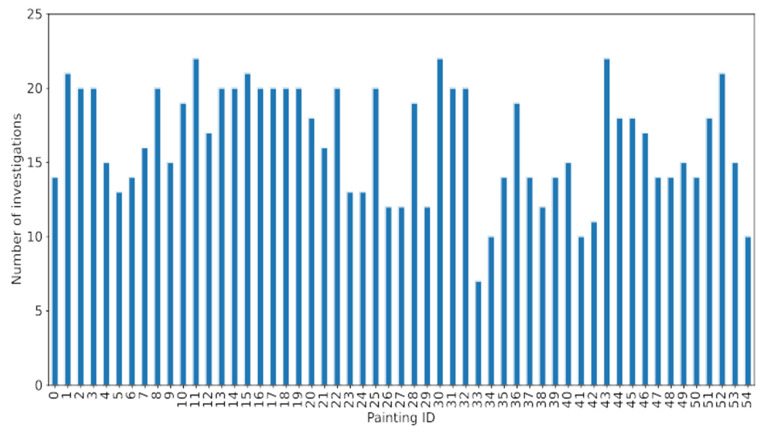
Number of investigations conducted for paintings in the dataset. The data does not contain any sample with all 32 markers collected. Usually, less than 20 tests were performed to characterize an item.

**Figure 8 molecules-27-00070-f008:**
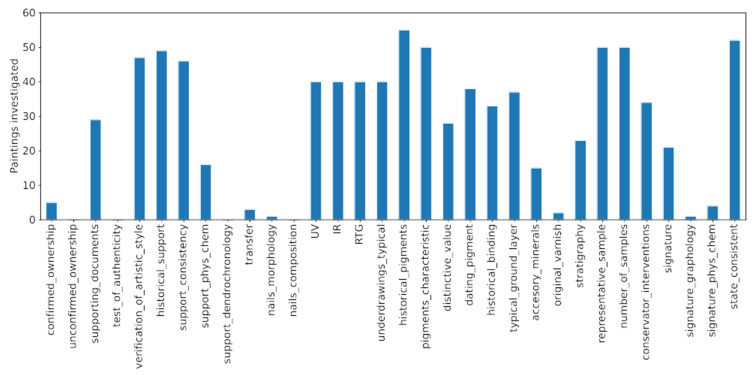
Frequencies of the usage of different attribution markers in the dataset. Five markers (unconfirmed ownership, test of authenticity, support dendrochronology, nails composition and original varnish) were not used at all and may probably be omitted in further authenticity investigations.

**Figure 9 molecules-27-00070-f009:**
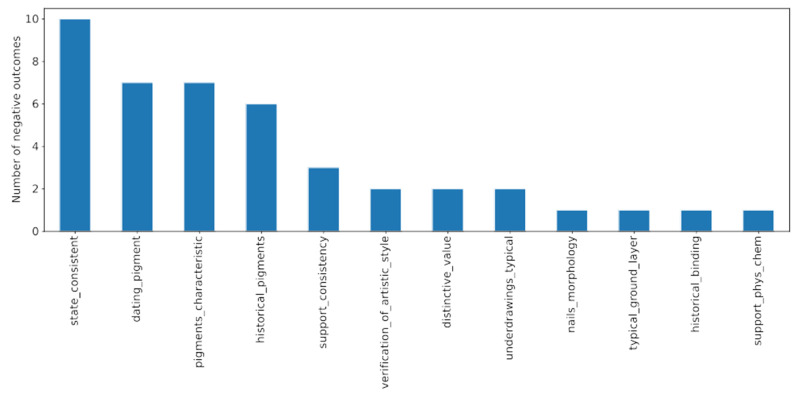
Investigations with negative outcomes. Since the forged paintings are expected to be characterized by at least one negative measurement, the features shown in this plot are of great importance for the examination process.

**Table 1 molecules-27-00070-t001:** List of proposed attribution markers. Every marker summarizes a series of measurements conducted on a given painting.

Marker	Marker Description
Confirmed Authorship	Ownership (or property rights) documents are in accordance with the law and verified by a lawyer
Unconfirmed Authorship	Ownership (or property rights) documents not verified by a lawyer
Supporting Documents	Documents supporting ownership or authorship (e.g., letters and photographs)
Test of Authenticity	Ownership, property rights and/or supporting documents verified by forensic investigations
Verification of Artistic Style	Art historian analysis confirming the style
Historical Support	Support consistent with the supposed time of the creation
Support Consistency	Support consistent with the painter techniques
Support Phys_Chem	Physicochemical examination of the support
Support Dendrochronology	Dendrochronological examination of the support
Transfer	Transfer (replacement) of the support
Nails Morphology	Historical analysis of nails’ morphology
Nails Composition	Physicochemical analysis of nails
UV	UV photography/UV luminescence
IR	IR photography/IR reflectography
RTG	X-ray photography/X-ray imaging
Underdrawings Typical	Underdrawings (or lack of them) consistent with the painter techniques
Historical Pigments	Pigments and dyes consistent with supposed time of creation
Pigments Characteristic	Pigments and dyes characteristic for the painter techniques
Distinctive Value	The characteristic feature of the painting techniques—the presence of the color underpainting
Dating Pigment	Dating pigment (i.e., pigment characteristic for the supposed time of creation)
Historical Binding	Binding media consistent with the supposed time of creation
Typical Ground	Ground layer typical for the painter techniques
Accessory Minerals	Physicochemical investigation of the primary layer and trace element analysis
Original Varnish	Presence of original varnish
Stratigraphy	Stratigraphy typical for painter’s techniques
Representative Sample	Samples representative for the object
Number of Samples/Number of Measurements Points	Sufficient number of samples/Number of Measurement Points
Conservator’s Interventions	Presence of conservator’s interventions
Signature	Signature attributed to the author
Signature Graphology	Handwriting investigations of the signature
Signature Phys_Chem	Physicochemical investigations of the signature
State Consistent	Declared state of the preservation consistent with the investigation results

**Table 2 molecules-27-00070-t002:** Our dataset consists of 55 paintings, each of which is characterized by a subset of attribution markers. An empty cell indicates that the corresponding marker was not investigated. Value “1” of a marker means that the corresponding examination was positive. Measurements with negative outcomes are represented by the value “−1”.

Marker	Painting ID																										
	0	1	2	3	4	5	6	7	8	9	10	11	12	13	14	15	16	17	18	19	20	21	22	23	24	25	26
confirmed_ownership																											1
unconfirmed_ ownership																											
supporting_ documents	1	1		1	1	1	1	1	1	1			1	1	1	1	1	1	1	1	1	1	1	1	1	1	
test_of_authenticity																											
verification_of_artistic_style	1	1		1	1	1	−1	1	1	1	1	1	1	1	1	1	1	1	1	1	1	1	1	1	1	1	1
historical_ support	1	1	1	1	1	1	1	1	1	1	1	1	1	1	1	1		1	1	1	1	1	1	1	1	1	1
support_c onsistency	1	1	1	1	1	1	1		−1	1	1	1	1	1	1	1	1	1	−1	1	1	1	1	1	1	1	1
support_phys_ chem	1				1	1	1			1	1		1				1				1			1	1		
support_dendrochronology																											
transfer											1								1								
nails_ morphology																											
nails_ composition																											
UV		1	1	1				1	1		1	1		1	1	1	1	1	1	1			1			1	
IR		1	1	1				1	1		1	1		1	1	1	1	1	1	1			1			1	
RTG		1	1	1				1	1		1	1		1	1	1	1	1	1	1			1			1	
underdrawings_typical		1	1	1	1		1	1	−1		1	1	1	1	1	1	1	1	1	1	1	1	1			1	1
historical_ pigments	1	1	1	1	1	1	1	1	1	1	1	1	1	1	1	1	1	1	1	1	1	1	1	−1	1	1	1
pigments_ characteristic	1	1	1	1	1	1	1		1	1	1	1	1	1	1	1	1	1	1	1	1	1	1	−1	1	1	1
distinctive_ value		1	1	1					1			1		1	1	1		1	1	1	1		1			1	
dating_ pigment	1	1	1		1	1	1	1	1	1	1	1	1			1	1		1	1	1	1		−1	1		1
historical_ binding	1	1	1	1		1			1	1	1	1	1	1	1	1	1	1	1	1			1	1	1	1	
typical_ground_layer		1	1	1				1	1			1		1	1	1	1	1	−1	1	1	1	1			1	
accesory_ minerals		1	1	1								1		1	1	1	1	1					1			1	
original_varnish																											
stratigraphy		1	1	1					1	1	1	1		1	1	1	1	1		1			1			1	
representative_ sample	1	1	1	1	1	1	1	1	1	1	1	1	1	1	1	1	1	1	1	1	1	1	1	1	1	1	1
number_of_ samples	1	1	1	1	1	1	1	1	1	1	1	1	1	1	1	1	1	1	1	1	1	1	1	1	1	1	1
conservator_ interventions	1	1	1	1	1		1	1	1		1	1	1	1	1	1	1	1	1	1	1	1	1			1	
signature			1		1			1		1		1	1								1	1					
signature_ graphology													1														
signature_ phys_chem												1									1	1					
state_consistent	1	1	1	1	1	1	1	1	1	1	1	1	1	1	1	1	1	1	1	1	1	1	1	−1	1	1	1
is_original?	YES	YES	YES	YES	YES	YES	YES	YES	YES	YES	YES	YES	YES	YES	YES	YES	YES	YES	YES	YES	YES	YES	YES	YES	YES	YES	YES

**Table 3 molecules-27-00070-t003:** Our dataset (continued). See the caption of Table 2 for more details.

Marker	Painting ID																										
	28	29	30	31	32	33	34	35	36	37	38	39	40	41	42	43	44	45	46	47	48	49	50	51	52	53	54
confirmed_ ownership	1	1											1														
unconfirmed_ ownership																											
supporting_ documents			1		1				1															1	1	1	
test_of_authenticity																											
verification_of_artistic_style	1		1	1	1		−1	1	1				1	1	1	1	1	1	1	1	1	1	1		1		1
historical_ support	1	1	1	1	1	1	1	1	1	1		1	1	1	1	1	1	1	1					1	1	1	1
support_c onsistency	1	1	1	1	1			1	1	1	1	1	1	1	−1	1	1	1						1	1	1	
support_phys_ chem		1							1						−1	1			1								
support_dendrochronology																											
transfer									1																		
nails_ morphology														−1													
nails_ composition																											
UV	1		1	1	1			1	1	1	1	1	1	1	1	1	1	1	1	1	1	1	1	1	1	1	1
IR	1		1	1	1			1	1	1	1	1	1	1	1	1	1	1	1	1	1	1	1	1	1	1	1
RTG	1		1	1	1			1	1	1	1	1	1	1	1	1	1	1	1	1	1	1	1	1	1	1	1
underdrawings_typical	1		1	1	1				1				1			1	1	1	1	1	1	1	1	1	1		−1
historical_ pigments	1	1	1	1	1	−1	−1	1	−1	1	−1	−1	1	1	1	1	1	1	1	1	1	1	1	1	1	1	1
pigments_ characteristic	1	1	1	1	1	−1	−1	1	−1	−1	−1	−1	1			1	1	1	1	1	1	1	1	1	1		
distinctive_ value	1		1	1	1			−1	−1								1	1	1	1	1	1	1		1		
dating_ pigment	1	1	1	1		−1	−1		−1	−1	−1	−1	1			1	1	1						1	1	1	
historical_ binding			1	1	1								−1		1	1	1	1						1	1	1	
typical_ground_layer	1		1	1	1				1	1		1	1	1	1	1	1	1	1	1	1		1	1	1		1
accesory_ minerals			1	1	1																				1		
original_varnish																1										1	
stratigraphy	1		1	1	1											1						1		1	1		
representative_ sample	1	1	1	1	1		1	1	1	1	1	1				1	1	1	1	1	1	1	1	1	1	1	
number_of_ samples	1	1	1	1	1		1	1	1	1	1	1				1	1	1	1	1	1	1	1	1	1	1	
conservator_ interventions	1		1	1	1											1			1	1		1	1	1	1	1	
signature		1	1			1	1	1		1	1	1				1			1	1	1	1					
signature_ graphology																											
signature_ phys_chem																1											
state_consistent	1	1	1	1	1	−1	−1	−1	−1	−1	−1	−1	−1			1	1	1	1		1	1	1	1	1	1	−1
is_original?	YES	YES	YES	YES	YES	YES	NO	NO	NO	NO	NO	NO	NO	NO	NO	YES	YES	YES	YES	YES	YES	YES	YES	YES	YES	NO	NO

**Table 4 molecules-27-00070-t004:** Relative importance of the features for the decision tree shown in Figure 5. Only the markers with nonvanishing importance are listed. The remaining ones can be neglected (they have no impact on the classification results).

Attribution Marker	Importance
State_consistent	0.864
Supporting_documents	0.081
Typical_ground_layer	0.055

**Table 5 molecules-27-00070-t005:** Accuracy of the classifier shown in Figure 5. The model was tested on the test data (paintings with IDs from 46 to 55 in Table 3). The predictions for the paintings are compared with their labels assessed by experts.

Painting ID	Predicted Class	Real Class
1	Authentic	Authentic
2	Authentic	Authentic
3	Authentic	Authentic
4	Authentic	Authentic
5	Authentic	Authentic
6	Authentic	Authentic
7	Authentic	Authentic
8	Forged	Forged
9	Authentic	Forged

**Table 6 molecules-27-00070-t006:** Relative importance of the features for the decision tree shown in Figure 6. The tree was built after the “state consistent” marker was removed from the data. Only markers with nonvanishing importance are listed. The remaining ones can be neglected (they have no impact on the classification results).

Attribution Marker	Importance
Pigments_characteristic	0.471
Distinctive_value	0.268
Representative_sample	0.132
Supporting_documents	0.128

## Data Availability

Not applicable.

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
