# Peer review of "Attribution Markers and Data Mining in Art Authentication"

_molecules, 2021, doi:10.3390/molecules27010070_

Round 1

Reviewer 1 Report

This paper presents Attribution Markers and Data Mining in the Art Authentication. However, there are several issues, For example,

1- no contribution

2- Old machine learning method has been used and only one what about other methods.

3- Code not attached and also the dataset

4-  no comparison with other methods.

5- no results from literature 

Reviewer 2 Report

General comments

This paper tried to apply decision trees to classify the authentication of paintings based on their several characteristics. Utilizing a machine learning method in the art authentication would be a good lesson for researchers in the field. However, there are several curiosities about if the authors have used the decision tree method in a right way.

Globally, I would like to recommend a major revision to address the curiosities before moving forward the next process. The detailed comments are below in "Specific comments".

Specific comments

Major comments

  1. In Figure 9, the decision tree used all predictors like a continuous variable, not a categorical variable, even though each predictor consists of -1, 0, and 1. I think they should be used with categorical variables like the predictors (Famous name, Genre, and Success) in the movie example in Table 3 and Figure 5.

  1. In "Data set characteristics" section, how did the authors split training set and test set? Randomly? I am curious if the same interpretation in "Relevance of the attribution markers" works, even though training data is changed.

  1. In "Classification of new paintings" section, my biggest concern is that the 100 percent accuracy was obtained in a specific training and test data. In machine learning studies, cross-validation scheme is necessary to evaluate the performance of a machine learning model more reliably. For decision trees, it is particularly required because it is well known that decision trees suffer from high-variance problems. I would also recommend a replication with a cross-validation, e.g. a stratified 5-fold cross validation with 100 replications.

  1. In the whole analysis of decision trees, did the authors use a pruning process? In Figure 9, all leaf nodes are pure. It suggests that the tree suffers from over-fitting problems to the training data. I also suggest that the tree cannot be generalized well to test data. It may be found if the decision trees are evaluated with a cross-validation scheme.

Minor comments

  1. In general, I feel like writing contains lots of unnecessary descriptions. The manuscript should be more clarified.

  1. All tables should share their formats equally to each other. Almost all Figures and Tables have a poor quality. It is too hard to understand the Figures and Tables because their descriptions are uninformative. E.g. Table 2 only contains six paintings out of 55 paintings? Figure 9 is too wide.

  1. In line 307-309, total number of paintings is 55, but sum of training set, 46, and test set, 10, is not 55.

Round 2

Reviewer 1 Report

Accept the paper, no more comments

Reviewer 2 Report

I would like to encourage the authors with their improvement. Most of my curiosities are addressed for now. The manuscript has a good shape for publication.